# An Efficient Humanized Mouse Model for Oral Anti-Retroviral Administration

**DOI:** 10.3390/cells12071034

**Published:** 2023-03-28

**Authors:** Amber K. Virdi, Sang Ho, Melanie S. Seaton, Arnold Z. Olali, Srinivas D. Narasipura, Hannah J. Barbian, Leannie J. Olivares, Hemil Gonzalez, Lee C. Winchester, Anthony T. Podany, Ryan D. Ross, Lena Al-Harthi, Jennillee Wallace

**Affiliations:** 1Department of Microbial Pathogens and Immunity, Rush University Medical Center, Chicago, IL 60612, USA; 2Center for Mitochondrial and Epigenomic Medicine, Children’s Hospital of Philadelphia, Philadelphia, PA 19104, USA; 3Department of Internal Medicine, Division of Infectious Diseases, Rush Medical College, Chicago, IL 60612, USA; 4UNMC Center for Drug Discovery, University of Nebraska Medical Center, Omaha, NE 68182, USA; 5Department of Anatomy & Cell Biology, Rush University Medical Center, Chicago, IL 60612, USA

**Keywords:** HIV, anti-retroviral, mouse model, latency, integration, human cell trafficking, viral load

## Abstract

HIV anti-retrovirals (ARVs) have vastly improved the life expectancy of people living with HIV (PLWH). However, toxic effects attributed to long-term ARV use also contribute to HIV-related co-morbidities such as heart disease, bone loss and HIV-associated neurocognitive disorders (HAND). Unfortunately, mouse models used to study the effects of ARVs on viral suppression, toxicity and HIV latency/tissue reservoirs have not been widely established. Here, we demonstrate an effective mouse model utilizing immune-compromised mice, reconstituted with infected human peripheral blood mononuclear cell (PBMCs). ARVs areincorporated into mouse chow and administered daily with combination ARV regimens includingAtripla (efavirenz, tenofovir disoproxil fumarate, and emtricitabine) and Triumeq (abacavir, dolutegravir and lamivudine). This model measures HIV-infected human cell trafficking, and ARV penetration throughout most relevant HIV organs and plasma, with a large amount of trafficking to the secondary lymphoid organs. Furthermore, the HIV viral load within each organ and the plasma was reduced in ARV treated vs. untreated control. Overall, we have demonstrated a mouse model that is relatively easy and affordable to establish and utilize to study ARVs’ effect on various tissues, including the co-morbid conditions associated with PLWH, such as HAND, and other toxic effects.

## 1. Introduction

Human immunodeficiency virus (HIV) is a retrovirus that integrates its genome into the host DNA and causes life-long infection. Combination anti-retrovirals (ARVs) have extended and improved the lives of people living with HIV (PLWH), by targeting multiple steps of the HIV replication cycle. However, ARVs are not a cure and PLWH must remain on therapy for the duration of their lives, as interruption leads to the resurgence of HIV replication. Although vital, long-term use of ARVs can cause toxicities that reduce the quality of life and are linked to several co-morbid conditions such as bone loss, neurocognitive deficits and heart disease [1,2,3,4,5,6,7]. The dynamics of ARV-induced toxicities and the extent of their impact, both independently and during HIV infection, are not fully understood. 

Commonly used mouse models to study the effects of ARVs in tissues have several caveats. These include: (1) drug availability: models predominately rely on ARV delivery through intraperitoneal (IP) injection or oral gavages, which do not adequately model the pharmacokinetics of human drug delivery. (2) Therapeutic tolerability: daily injection or gavage induces significant stress in the animals, which compounds adverse health consequences. (3) ARV solubility: only a limited number of ARVs are water soluble and the injections/gavages require additional vehicles such as dimethyl sulfoxide (DMSO), which can be toxic on its own [8]. These caveats introduce variables that detract from the validity of these models, and create more potential for systemic adverse effects in the animal. Currently, small animal models of HIV infection and ARV administration do not ade-quately recapitulatethe routes of transmission and drug administration in humans [9]. Additionally, ARV concentrations in various HIV-relevant tissues are poorly defined in models relying on injection/gavage. 

One already established, cost-effective small animal model of HIV infection is the hu-PBMC NOD-SCID IL2Rgamma^null^ (hu-PBMC-NSG) mouse. This immunocompromised model does not require the surgical transplantation of human tissue for humanization, but rather a simple IP injection of mononuclear cells [10]. Our laboratory has previously demonstrated that the brain tissue of hu-PBMC-NSG mice is reconstituted with PBMCs, and that HIV infection can occur in this important tissue site. Not only that, our lab has shown that the virus can egress from the brain and infect the periphery, reasserting that the brain is an extremely important organ to assess in HIV infection [11]. As these studies utilized IP-injected ARVs, we have set out to create a more physiologically relevant way to expand upon this model. Our objective in this study is to develop a cost-effective and accessible mouse model for ARV administration that can be utilized for assessing HIV infection, latency, tissue reservoirs and subsequent neuro-inflammation. Here, we describe the dynamics of humanization, infection and ARV concentration in hu-PBMC-NSG mice after acute oral administration of a therapeutic via animal chow. 

In this study, the combination ARVs used were Atripla (tenofovir disoproxil fumarate (TDF), emtricitabine (FTC) and efavirenz (EFV)) and Triumeq (abacavir (ABC), lamivudine (3TC) and dolutegravir (DTG)). The drugs were added in the facility-provided chow at dosages within the range of previous studies. We detected both ARVs, specifically: EFV, DTG, TDF and ABC, as well as PBMCs in the brain, liver, lungs, kidney, lymph nodes and distal and proximal colon. Furthermore, the combination Triumeq successfully reduced the plasma and tissue viral load, and while 3TC was not detected in tissues, it was found in plasma. As studies continue to elucidate the effects of HIV infection and long-term combination ARV therapy, it is important to have access to in vivo models that can recapitulate the relevant biological features of infection and ARV use. Humanized mouse models such as the one we describe in this study can be used as an efficient and easily attainable tool for studying infection as well as ARVs’ effects, both in combination or individually.

## 2. Materials and Methods

### 2.1. Ethics Statement

Research involving humans and animals was conducted in accordance with the institutional and U.S. government guidelines on human (IRBL06080703-CR13) and animal research (IACUC 20-018). Whole blood was collected from a healthy seronegative donor at Rush University Medical Center, with signed informed consent prior to donating.

### 2.2. Anti-Retroviral Drugs

The combination drug therapeutics consisted of Atripla (tenofovir disoproxil fumarate (TDF), emtricitabine (FTC) and efavirenz (EFV)) and Triumeq (abacavir (ABC), lamivudine (3TC) and dolutegravir (DTG)). Concentrations were chosen to be within the wide range of those reported in the literature. The concentrations used were 8.57 mg/kg (ABC), 0.71 mg/kg (DTG), 4.28 mg/kg (3TC), 8.57 mg/kg (EFV), 2.85 mg/kg (FTC) and 4.28 mg/kg (TDF). Mouse chow and ARVs at the aforementioned concentrations were measured out for 4 weeks of consumption. Since an average mouse weighing within 25–30 g will consume between 3 g and 6 g of food per day [12], the mouse chow used was rehydrated with sterile water and thoroughly mixed with powdered ARV drugs into a homogenous slurry, and formed into pellets weighing 4–6 g using molds. The ARV chow was left to air dry overnight. Optionally, resting pellets in a well-ventilated area may improve drying. The ARV chow was stored at 4 °C for short-term use, or at −20 °C for periods longer than 4 weeks. One to three days’ worth of ARV chow pellets was added to cages containing groups of five mice (i.e., five to fifteen ARV chow pellets for one to three days, respectively). 

### 2.3. Mouse Models, Humanization and HIV Infection

C57BL/6 male mice were administered the two different ARV combinations for 4 weeks to determine tissue distribution of each of the six drugs. In order to determine whether the ARV chow would reduce the HIV viral load, hu-PBMC-NSG mice were used and treated with the 3TC/ABC/DTG (Triumeq) ARV combination only. For the hu-PBMC-NSG model, peripheral blood mononuclear cells (PBMCs) from a healthy male donor were isolated using Ficoll-Hypaque density gradient centrifugation [10]. Cells were cultured in complete RPMI 1640 medium (ThermoFisher, Waltham, MA, USA), which included 10% heat-inactivated fetal bovine serum (Gemini Bio Products, Calabasas, CA, USA), 1% penicillin/streptomycin (Sigma-Aldrich, St. Louis, MO, USA), 2 mM L-glutamine (GlutaMAX supplement; ThermoFisher, Waltham, MA, USA) and 20 U/mL IL-2 (AIDS Reagent Program, Germantown, MD, USA). Cells were activated with 1 ug/mL of purified NA/LE mouse anti-human CD3 clone OKT3 (BD Biosciences, Franklin Lakes, NJ, USA) and 1 ug/mL NA/LE mouse anti-human CD28 clone CD28.2 (BD Biosciences, Franklin Lakes, NJ, USA). The cells were then infected with HIV_BaL_ at 2 ng/10^6^ cells. PBMCs were required to express at least 6–10% intracellular p24^+^ before reconstitution of mice; typically necessitating 4–6 days of infection. NOD-scid IL2Rgamma^null^ (NSG) mice were obtained from Jackson laboratory (Bar Harbor, ME, USA; no. 00557) aged 5–7 weeks. Mice were reconstituted via intraperitoneal (IP) injection with approximately 10 million cells per mouse once the infection threshold was achieved. Reconstituted mice were then left to humanize for 1 week before commencing ARV therapy via chow pellets for 4 weeks (Figure 1A). 

### 2.4. Mouse-Tissue Harvesting

After 4 weeks of ARV therapy, mice were euthanized by CO_2_ inhalation, followed by cardiac perfusion with 30 mL ice-cold sterile DPBS (Corning, Corning, NY, USA) through the left ventricle. Harvested tissues, including spleen, lungs, liver, kidney, heart, brain and distal and proximal colon tissue were placed in ice-cold PBS, and the peripheral blood was collected in EDTA coated tubes to be used for further assessment.

### 2.5. HIV Measurements

After harvesting tissues from control and ARV-treated mice, tissue sections were cut, weighed and stored in RNALater Stabilizing Solution (ThermoFisher, Waltham, MA, USA) until use. Before vRNA and vDNA isolation, weighed tissue sections were homogenized with ceramic beads for hard and soft tissue (VWR, Radnor, PA, USA) at 4000 rpm for 3 min using the Benchmark Bead Blaster 24 (Benchmark Products, Lincolnville, IL, USA). RNA was isolated the per manufacturer’s instructions using an RNeasy kit (Qiagen, Valencia, CA, USA), and DNA was isolated per the manufacturer’s instructions using a DNAeasy kit (Qiagen, Valencia, CA, USA) per the manufacturer’s instruction. The nucleic acids were quantified using a nanodrop (ThermoFisher, Waltham, MA, USA).

### 2.6. Plasma Viral Load

Following blood collection from the ARV-treated and untreated control HIV-infected hu-PBMC-NSG mice, the plasma was collected via the centrifugation of mouse blood for 15 min at 7000 rpm. Plasma viral RNA was isolated using a QIAamp Viral RNA Mini Kit (Qiagen, Valencia, CA, USA) per the manufacturer’s instructions. An RNA Quantification Standard ARP-3443 [13] was utilized for the determination of the plasma viral load. After isolation, 140 uL of the template was used in q-RT-PCR using a qScript cDNA Synthesis (Quanta Biosciences, Gaithersburg, MD, USA), followed by nested PCR, as previously described [11].

### 2.7. Flow Cytometry

Cells in suspension were stained for extracellular markers CD3, CD4 and CD8 diluted in a stain buffer containing FBS (BD Biosciences, San Jose, CA, USA) for 30 min at 4 °C, then fixed and permeabilized with Cytofix/Cytoperm (BD Biosciences, San Jose, CA, USA) for 30 min at 4 °C. Intracellular staining with p24 antibody (HIV-1 core antigen-RD1, KC57; Beckman Coulter, Brea, CA, USA) was performed from 1 h to overnight at 4 °C. The samples were assessed on the BD LSRFortessa (BD Biosciences, San Jose, CA, USA). The flow cytometry data collection and analysis were conducted using FACSDiva software (BD Biosciences, San Jose, CA, USA) and FlowJo software v10.7.1(TreeStar, Ashland, OR, USA), respectively.

### 2.8. Tissue Viral Load

The HIV RNA and DNA copy numbers were quantified by generating a standard curve. A 500 bp gBlock gene fragment encompassing the gag region [11] was synthesized (Integrated DNA Technologies, Coralville, IA, USA). The copy number was calculated using the DNA copy number web tool (ThermoFisher, Waltham, MA, USA) and converted to an RNA copy number by multiplying by 2. The fragment was suspended in sterile TE buffer at dilutions ranging from 10 to 10^6^ copies/μL, aliquoted and stored at −80 °C. The RNA extracted from the tissues was quantified using a nanodrop and approximately 500 ng total RNA was converted into cDNA using Qscript cDNA synthesis (Quanta Biosciences, Gaithersburg, MD, USA). The assay was performed with the Qultraplex 1-Step Toughmix Low (Quanta Biosciences, Gaithersburg, MD, USA) on the QuantStudios 5. The PCR conditions were 50 °C for 10 min and 95 °C for 3 min, followed by 40 cycles of 95 °C for 3 s and 55 °C for 30 s. The sequences of primers and probes used were as follows: Gag Probe FAM, /55-FAM/TTCGCAGTC/ZEM/AAT; Gag Forward Primer, GCAAGCAGGGAACTAGAAAGA; Gag Reverse Primer, CTGTCTGAAGGGATGGTTGTAG. The amount of tissue processed, total volume of nucleic acids extracted and dilutions performed at each step (cDNA synthesis, qPCR set up etc.) were considered in the calculations to finally equate to the copy number of viral nucleic acids per mg of tissue.

### 2.9. Human DNA Quantification in Tissue

Highly specific human DNA was quantified in the tissue via Taqman qPCR in all the isolated tissue DNA to determine human cell trafficking in animals. Approximately 500 ng of total DNA was used and the assay was performed with the Qultraplex 1-Step Toughmix Low (Quantabio, Beverly, MA, USA) on the QuantStudios 5. The PCR conditions were 50 °C for 10 min and 95 °C for 3 min, followed by 40 cycles of 95 °C for 3 s and 55 °C for 30 s. The sequences of primers and probes used were as follows: 

human DNA GFAP forward, ACCCAGCAACTCCAACTAAC;

human DNA GFAP reverse, TTCTCTCCTTCCTCCTCATTCT;

human DNA GFAP probe, /56-FAM/CATGGCCAG/ZEN/CAGCTTGCGTT/3IABkFQ/;

mouse DNA GFAP forward, CTGAACTTAGCCCTCCACAG;

mouse DNA GFAP reverse, TCACCTCCTCATAGATCTTCCT;

mouse DNA GFAP probe, /5FAM/TGGCTCGTG/ZEN/TGGATTTGGAGAGAA/3IABkFQ/.

A standard curve was created using a synthesized gblock for a 500 bp gene fragment encompassing the target sequence as mentioned above, which was serially diluted prior to qPCR. The DNA copy number per mg of tissue was then expressed in cell number, assuming 2 copies of DNA per cell.

### 2.10. Mass Spectrometry

Brain, heart, kidneys, liver, lungs, spleen, proximal and distant colon and plasma (for infected hu-PBMC-NSG mice only) were weighed and subsequently homogenized in 0.5 mL of 70% methanol. The tissues were homogenized with a Precellys Evolution Cryolys homogenizer (Bertin Technologies, Rockville, MD, USA) in a temperature-controlled chamber (−20 °C) according to the manufacturer’s protocol and as described previously [14]. Tissue homogenates were centrifuged at 10,000 rpm for 20 min and supernatants containing drugs were collected into fresh tubes and stored at −80 °C until used for drug quantification. Quantitation of plasma and cell associated DTG, TFV, FTC, TFV-diphosphate (TFV-DP; TDF intracellular anabolite) and FTC-triphosphate (FTC-TP; FTC intracellular anabolite) concentrations was performed with validated LC/MS/MS methods as described previously [15,16,17].

## 3. Results

### 3.1. Human PBMC Trafficking to the Peripheral Tissues Correlates with the Viral Burden

hu-PBMC-NSG model utilizes infected human PBMCS, as verified via the intracellular p24 flow cytometry, reconstituted into immunocompromised animals for 7 days prior to ARV initiation (Figure 1A). The animal weight was not significantly different between the treatment groups over the treatment time, nor were there significant differences in weight over time (Figure 1B). The human DNA content was quantified among all target tissue types in the human-PBMC-NSG mice via qRT-PCR. Using a standard curve, we quantified the copies of human DNA per mg of tissue and represented it as a human-to-mouse cell ratio (the number of human cells divided by the number of mouse cells). When comparing across the harvested tissue, we observed that the tissue associated with lymphocyte circulation and animal humanization, such as the spleen and lymph nodes, contained the highest levels of human DNA, as expected (Figure 1C). Additionally, all the assessed organs showed signs of human cell reconstitution including the lungs and colon, as well as the heart and the brain. A comparison of this ratio to the HIV DNA copy numbers in tissues indicated a positive linear correlation (Figure 1D), suggesting higher immune cell trafficking results in a higher level of viral integration. The same trend was seen when the human cell number was correlated to the HIV RNA copy number, indicating a similar trend for the viral replication (Figure 1E). As expected, the lymph nodes, lungs and spleen, all of which are secondary lymphoid organs, show the highest level of human to mouse DNA ratio, although all tissues show evidence of human cell content.
Figure 1Human PBMCs trafficking to the peripheral tissues correlates with viral burden. (**A**) Immunocompromised mice were reconstituted with PBMCs that had been infected with HIV for 7 days, prior to commencement of ARV therapeutic. The animal weight did not differ between the groups over the treatment time (**B**). In order to determine if the human cell trafficking was associated with the viral load in the peripheral tissue, the human DNA was quantified and expressed as a cell ratio of mouse DNA within each tissue type, providing relative human DNA content in every tissue (**C**). This was then correlated with (**D**) the viral DNA and (**E**) the viral RNA to determine if the human cell trafficking correlates with the viral burden.
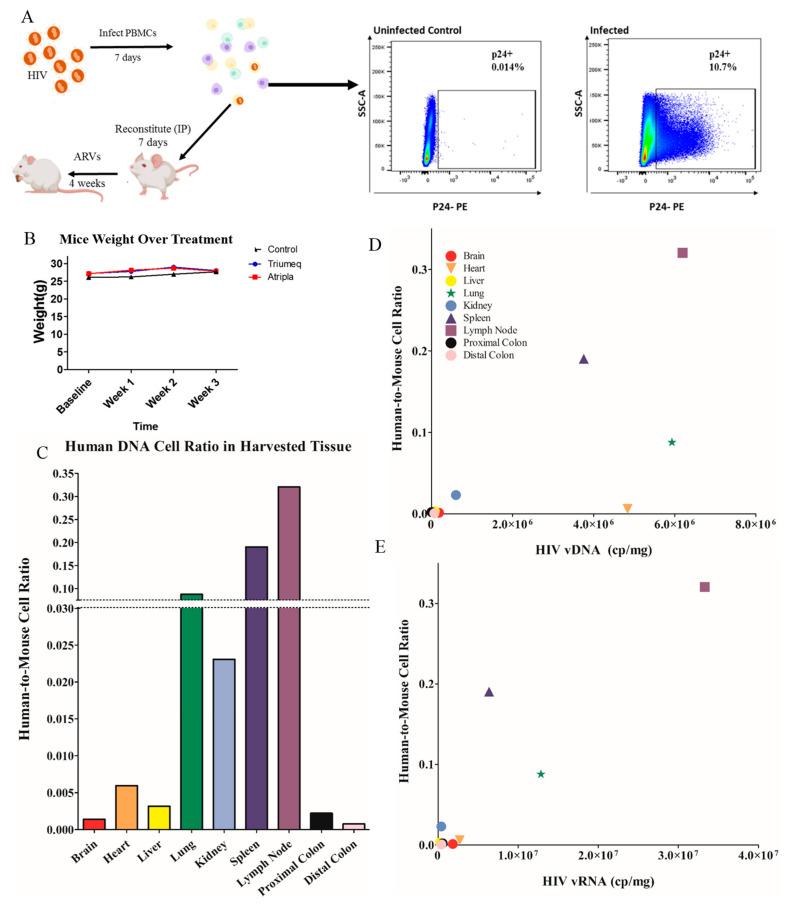


### 3.2. ARVs Were Detected in All Assessed Tissues

To determine the ARV penetration into various organs, mass spectrophotometry was performed on C57BL/6 mice administered ARV chow for 4 weeks. Of the tissues, the colon and heart had the highest overall penetration, while the brain had the lowest (Figure 2). EFV, TDF, DTG and ABC were found in all tissues. 3TC and FTC were not detected in tissues; however, we measured 3TC in the plasma of animals that received the 3TC/ABC/DTG combination (Triumeq) (Figure 2). We assessed the plasma ARV levels in the Triumeq-receiving mice, since Triumeq is currently more widely prescribed than Atripla due to the documented EFV and TDF toxicities [18], and therefore it was the ARV combination we went on to use to treat infected hu-PBMC-NSG mice. DTG had the highest overall concentration across all the tissues, despite having the lowest administered dose out of all ARVs (0.71 mg/kg) (Figure 2). This data set demonstrates that oral administration of ARVs via mouse chow results in detectable ARVs in several tissues.

### 3.3. ARV Administered through Chow Effectively Suppresses HIV in the Tissue Targets

Next, we assessed the ability of oral ARV administration to inhibit HIV infection in the hu-PBMC-NSG infected mice. All tissue sites contained less vRNA (Figure 3A) and vDNA (Figure 3B) in ARV chow-fed animals when compared to untreated, demonstrating that the drugs that penetrated tissues were able to inhibit the HIV replication and reduce the overall HIV burden. Notably, the heart, liver, lungs and lymph nodes showed the greatest reduction in vRNA, while the distal and proximal colon showed the least (Figure 3A). The lymph nodes, heart and lungs showed the highest reduction in vDNA content, while the brain and both colons showed the lowest (Figure 3B). 

### 3.4. ARV Administration Skews Infected Cells towards Latency

Finally, the HIV RNA/DNA ratio was assessed for each tissue type. This allows for the correlation of transcription and integration to determine the establishment of latent reservoirs and persistence over ARV treatment. The highest RNA to DNA ratio was in the brain, liver, spleen and proximal colon (Figure 4). Interestingly, the lymph nodes show an increase in vRNA to vDNA content with ARV treatment, demonstrating more transcription. 

## 4. Discussion

Within this study, we have established a physiologically relevant and cost-effective humanized mouse model that can be used to assess the HIV infection and oral ARV administration. Most humanized mouse models that assess HIV infection and treatment include the use of single drugs administered via IP injection; however, this approach does not recapitulate the current clinical treatment regimens. To our knowledge, our model is among a few currently being used to assess oral combination therapies. Other models of oral ARV administration use humanization protocols that introduce variables prior to infection and treatment, such BLT mice that require the surgical implantation of human tissue [19] or a huCD34^+^ NSG model that requires the reconstitution of neonatal animals [20]. Both of these models arevery useful researchtools,they are relativelytime intensive,techni-cally demanding, andexpensive and pose the risk for undue stress to the animal. Our hu-PBMC-NSG model provides an alternative approach that allows for the immediate use of the animals, without the procurement of human tissue or hematopoietic stem cells, which are difficult to obtain due to low accessibility and cost. Though one should note that this model has little to no granulocytes. Here, we assessed the distribution of infected human cells and determined the ARV dissemination and the impact on infection within the therapeutically relevant tissues in our model.

To begin characterizing our model, we assessed the distribution of human cells across tissues. We utilized a highly specific Taqman RT-PCR to determine a copy number of human DNA, and ultimately an estimation of number of cells per mg of tissue in only HIV^+^ untreated mice. The average human cell number was proportioned to the mouse cell number in the same organ, and correlated with vDNA and vRNA content of the tissue type (Figure 1). Interestingly, the tissues with the highest human DNA content (Figure 1D) were associated with secondary lymphoid organs; the spleen, lung and lymph nodes. These organs would see a high human cell trafficking in the animals, especially the spleen where an enlarged spleen size is a physical indicator of the successful engraftment of human cells and reasserting the physiological relevance of this model. Data showed trending positive correlations between the human DNA content and vDNA load, where tissues that had higher human cell counts also showed a higher viral content during untreated HIV (Figure 1D). This trend persisted compared to vRNA (Figure 1E), suggesting that the viral burden is correlated with the cell dissociation within the peripheral organs, as expected. Overall, our data suggest in the hu-PBMC-NSG model that the human DNA content is highest in lymphatic sites and is correlated with the viral burden, which mirrors what would be expected in a physiologically relevant model of infection. 

When assessing this model further, we chose to focus on the secondary lymphoid organs such as the spleen, lung and lymph nodes, as well as the sites of HIV-associated comorbidities such as the gut, heart and brain. Across all tissues, we report an expected decrease in viral load with ARV therapeutics; the heart, lungs and lymph nodes show the most reduction in both vRNA and vDNA, while the proximal and distal colon shows the least. The brain did not show elevated levels of vRNA relative to other tissue pre-treatment; however, ARVs decreased both the vRNA and vDNA in this compartment. The lymph nodes and lungs are considered to be secondary lymphoid organs and as HIV primarily targets human CD4^+^ T cells within this mouse model, the most viral load reduction we would see would be in these high CD4^+^ trafficking organs. The colon had one of the highest levels of ARV penetration of all the organs (Figure 2), yet both vRNA and vDNA show the least amount of viral reduction with ARV therapeutics (Figure 3). This may be partially attributed to the colon being “isolated” due to the tight control of leukocyte trafficking to the colon [21]. This is supported by Figure 1, in which we see the proximal and the distal colon exhibit minimal human cell trafficking. It is important to note that due to humanization, there is some animal-to-animal variability that exists in this model. While there was no difference in PBMC donors, some tissue compartments showed a higher variability in vRNA and vDNA levels (Figure 3) and this may be due to the expected variation in viral dissemination and ARV administration that occurs in animal models. In future experimentation, a higher animal number should be sufficient to address these variations.

While not the objective of this paper, this model could be utilized to further assess the dynamics of HIV latency. A ratio of transcriptional activity (vRNA) over latency (vDNA) could indicate whether ARV treatment impacts the infection dynamic within each tissue type (Figure 4). We took this ratio in animals that were both treated and untreated with ARVs and we observed that eight out of the nine organs have repressed transcriptional activity in Triumeq-treated mice. The increase in vRNA/vDNA ratio seen in the lymph nodes in Figure 4 denotes that the ARV-treated animals have more transcriptional activity compared to untreated, which sets this organ apart from all the other harvested tissue. While unexpected, it is important to reiterate that in this animal model, the lymph nodes are only present after humanization, therefore they comprised almost exclusively human cells, hence their high human cell trafficking levels (Figure 1B). This dynamic could be the explanation for the lymph nodes being outliers in vRNA/vDNA ratio, as they have less host influence than any other tissue set. This further use of the model could also prove extremely useful in determining the role of tissue reservoirs in HIV infection on and off treatment, especially in the brain. These results indicate that ARV exhibited high efficacy in reducing the overall viral load, showcasing the effectiveness of our model.

Despite the longer lifespans of PLWH due to use of ARVs, there are still accompanying toxicities that demand more insight and research into the dynamics of the drugs themselves. Considering the rapid impact of ARVs on the viral burden after initiation, there is a known dissemination of the drug throughout the body within the first couple of months of therapeutics. Here, we have outlined an acute ARV chow model in hu-PBMC-NSG animals that shows the peripheral circulation of human cells, HIV virus and ARV drugs, while effectively modeling viral suppression. Due to the accessibility of this model, its use in furthering the understanding of ARVs can extend beyond the context of HIV infection and into pre-exposure prophylaxis (PREP) models, which are few and far between, as there is a need for the methodical and rapid testing of this therapeutic context [14,22]. In addition, models to study HAND have not been fully realized and our model may prove to be a viable one and our model may be utilized in experiments pertaining to viral rebound studies, behavioral studies and long-term latency models. In this paper, we have outlined an efficient and cost-effective mouse model to study the use of ARVs; however, there comes a few caveats with our model. Our model shows human cells traffic to HIV relevant organs, with secondary lymphoid tissue and the heart having the highest infiltration. One limitation may be the need to extend the timeline for humanization so as to allow longer time for dissemination. That being said, our model is prone to graft-versus-host-disease (GVHD) if the animal is humanized and left for longer than 8 weeks. This limits the longevity of the experimental timeline so that there are no additional variables introduced through GVHD. Lastly, based off our drug penetration data, ARV components such as FTC may have some difficulty reaching all the peripheral tissues. Despite these limitations, there is still a pressing need in the field for a mouse model that is assessable to all labs and could be used for multi-organ analysis.

In summary, we present an effective and affordable humanized mouse model that provides a more efficient and accessible way to deliver ARVs orally in HIV-infected mice. We illustrated that the oral administration of ARVs results in effective drug penetration into the tissues of importance and a reduction in the viral load in those tissues. We believe that our model can be used further in studies regarding HIV infection and immune dynamics in important tissue targets.

## Figures and Tables

**Figure 2 cells-12-01034-f002:**
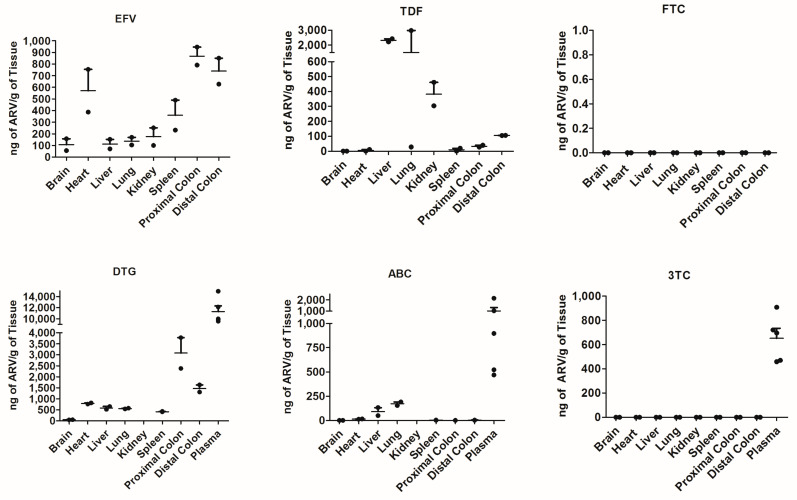
ARV chow delivers drugs into the peripheral tissue. The concentration of the individual ARVs measured by mass spectrophotometry in the brain, liver, plasma, spleen, kidneys, proximal and distal colon, lungs and heart of C57/BL6 mice that were fed ARV pellets every day for 4 weeks. Presented as ng of ARV/g of tissue. *N* = 2 mice. EFV: Efavirenz, DTG: Dolutegravir, TDF: Tenofovir disoproxil fumarate, ABC: Abacavir, 3TC: Lamivudine, FTC: Emtricitabine. Error bars show mean ± SEM.

**Figure 3 cells-12-01034-f003:**
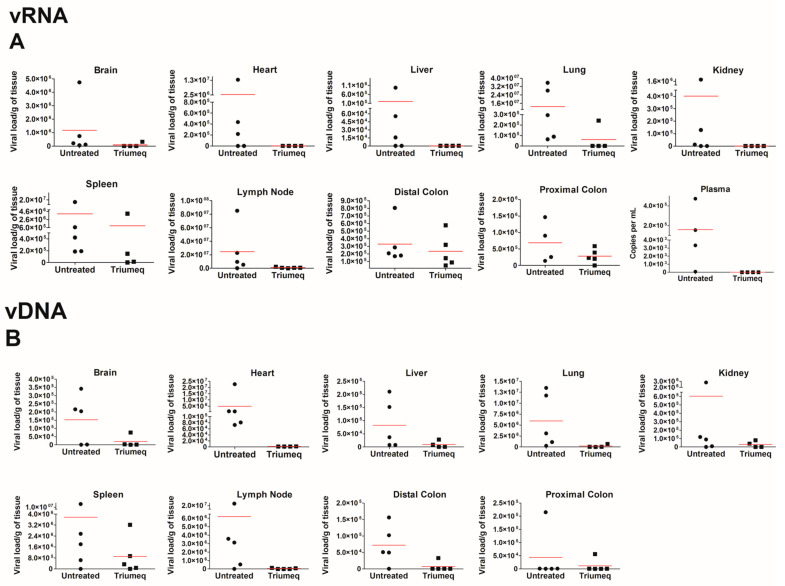
ARV chow reduces the HIV RNA and DNA in peripheral tissues. Brain, heart, liver, lungs, kidneys, spleen, lymph nodes, proximal colon, distal colon and plasma from HIV-infected hu-PBMC-NSG mice treated with ARVs for 4 weeks were isolated and the viral load was determined from (**A**) the isolated RNA and normalized to grams of tissue. The undetermined values were set to a CT of 45. (**B**) The vDNA isolated from different tissues and normalized similarly. Untreated mice are represented by closed circles ●, while Triumeq treated are represented by closed squares ■.

**Figure 4 cells-12-01034-f004:**
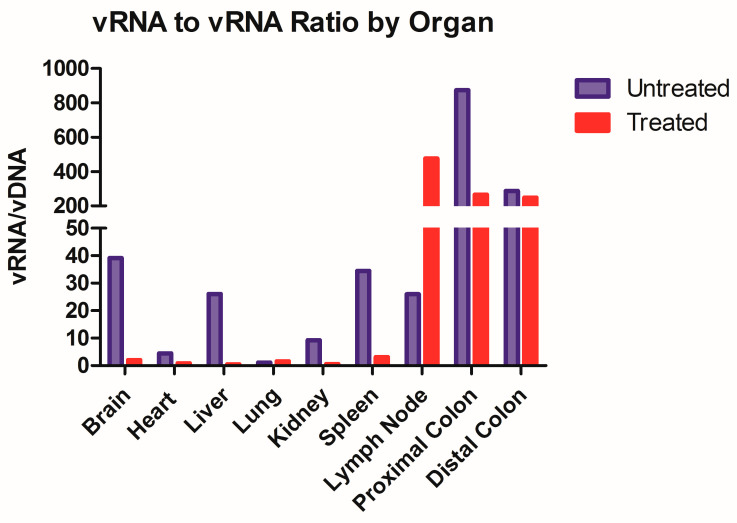
ARV initiation skews the vRNA/vDNA ratio towards a latent infection. The RNA viral load was divided by DNA viral load to calculate a ratio for each organ in untreated mice and treated mice, where an ARV therapeutic ratio less than the untreated signaled a skew towards HIV latency. All tissues showed this trend, with the exception of the lymph nodes, where ARV therapeutics increased the ratio. The undetermined values were set to a CT of 45.

## Data Availability

(1) All data supporting the finding of this study are available within the paper. (2) Any additional information required to reanalyze the data reported in this paper is available from the lead contact upon request.

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
