# Peer review of "An Efficient Humanized Mouse Model for Oral Anti-Retroviral Administration"

_cells, 2023, doi:10.3390/cells12071034_

Round 1

Reviewer 1 Report

Overall the manuscript is significant and will be of interest - the authors have proposed a useful model and it provides a necessary advancement/tool for the field.

There is room for improvement in few areas:

1. Methods elaboratoin:

The methodology regarding the ARV food preparation is lacking in detail. The reproducibility of the formulation will be an essential component if this will be a uniform model that others can implement.

What sex were the animals? What sex were the PBMC donors?

2. The discussion could be expanded to address:

The individual animals showed wide differences in viral RNA in both treated and untreated arms. Was this due to different donors of the human cells? What role did individual animal ARV levels play (comparing organs and plasma in figures 2 and 3 do not show a clear pattern).

Limitations of the model (ex. how long can the experiments be run?) should be discussed along with the future uses.

Data and Results:

The labels on the axes are much too small - can't be read even at 180%, and unclear beyond that.

The animals were group housed, which is important for lowering stress levels, but that makes it hard to assess food consumption. If daily weight data are available (or weekly), that lends insight into food consumption.

Did you find and ARV resistance mutations? Can you look for those and report on them? four weeks might be a bit short, but that would definitely be of interest in the field (to have a model to study ARV resistance development in different organs).

Author Response

Reviewer 1 

Overall the manuscript is significant and will be of interest - the authors have proposed a useful model and it provides a necessary advancement/tool for the field. There is room for improvement in few areas: 

  1. Methods elaboration:  

Comment #1: The methodology regarding the ARV food preparation is lacking in detail. The reproducibility of the formulation will be an essential component if this is a uniform model that others can implement.   

Reply #1: We thank the reviewer for this suggestion. The method for ARV pellet preparation has been updated on pages 2-3, highlighted in gray to include more detail pertaining to ARV food preparation.

Comment #2: What sex were the animals? What sex were the PBMC donors?   

Reply #2: We have updated the methods section accordingly as per your suggestion (highlighted in gray on page 3). All the mice were male, and we only had one also male healthy, non-smoking PBMC donor. We utilized one healthy non-smoking male donor and treated the PBMCs as a constant, so as to avoid/minimize within groups and between groups variability outside of the actual variable being assessed.      

  1. The discussion could be expanded to address:  

Comment #3: The individual animals showed wide differences in viral RNA in both treated and untreated arms. Was this due to different donors of the human cells? What role did individual animal ARV levels play (comparing organs and plasma in figures 2 and 3 do not show a clear pattern).  

Reply #3: Thank you for this suggestion, we have added a section to our discussion addressing the variation in viral load; page 10 highlighted in gray, beginning and ending with: “It is important to note...address these variations”.  

There was no difference in donor for animals in Figure 3, therefore the PBMCs used did not vary. As described in the manuscript, Figure 2 assessed the drug concentrations in tissue compartments in C57/BL6 mice while Figure 3 assessed viral suppression in humanized hu-PBMC-NSG animals.

Additionally, the ARV penetration and viral load burden in each tissue will not directly correlate because of: 1) tissue-specific pharmacodynamics and pharmacokinetics, and 2) the breadth of the immune cell compartment (specifically HIV target cells) varies significantly among tissues. Therefore, viral burden will also vary significantly across tissues.

Comment #4: Limitations of the model (ex. how long can the experiments be run?) should be discussed along with the future uses.   

Reply #4: Thank you for your suggestion and we agree. We have added a section to address the limitations of our model along with the future directions, highlighted in gray pages 10-11, beginning and ending with “In this paper...multi-organ analysis”.  

Data and Results:  

Comment #5: The labels on the axes are much too small - can't be read even at 180%, and unclear beyond that.  

Reply #5: We apologize for this and we have adjusted all figures to improve visibility.   

Comment #6: The animals were group housed, which is important for lowering stress levels, but that makes it hard to assess food consumption. If daily weight data are available (or weekly), that lends insight into food consumption.  

Comment #6: Thank you for this suggestion, yes we have weekly weight data. We did not observe significant changes which is why it was not initially presented. However, we agree with the reviewer that this is important data giving insight into food consumption. It is now illustrated as Figure 1B. 

Comment #7: Did you find any ARV resistance mutations? Can you look for those and report on them? Four weeks might be a bit short, but that would definitely be of interest in the field (to have a model to study ARV resistance development in different organs).  

Reply #7: We agree that a model of ARV resistance would be invaluable for the field. In these animals we did not CHECK because as the reviewer mentioned, 4 weeks is a shorter time period for this purpose and it was not an initial objective in this preparation. We do have the ability to expand our timeline by several weeks making this a potential characteristic to explore in future studies.

Reviewer 2 Report

This is a good manuscript, but requires major revision, as per comments addressed to the authors (attached document)

Author Response

Reviewer 2 

The manuscript of Sang et al describe and present data on transferring HIV to the brain of immune deficient mice by using PBMCs for the purpose of testing the retro-viral therapy. The manuscript is well written, though there are several concerns the authors need to address. 

Comment #1: Explain the non-inflammatory circumstances under which the PBMC can cross the BBB into the brain;  

Reply #1: The brain is not thought to be immune privileged as it once was, and there is evidence to suggest lymphatic movement in the CNS to maintain homeostasis, such as clearing debris from dying cells and immune surveillance. This was first conceptualized in the early 1900’s by Italian neuropathologists but was more recently elaborated upon when Louveau et al. eloquently described a system of lymphatics connecting the CNS and periphery, indicating routine immune cell trafficking between these two compartments1,2.

Comment #2: In the statistic section it is mentioned about using error bars +/- SEM for data in their graphs. However, the p values need to be shown;   

Reply #2: Thank you for the comment, we have removed that section due to the presentation of this data not necessitating these values. The scope of our manuscript is focused on describing this model and not comparing the dissemination of ARVs within the tissue compartments

Comment #3: It would be good the qRT-PCR data are doubled by immunofluorescence microscopy;   

Reply #3: Thank you for this suggestion and we agree that histology would be accompaniment useful addition, however much of our tissue was used to perform the assays in the study so that we could achieve sufficient outputs. We do plan on incorporating histology into future use of this model.  

Comment #4: In the discussion section authors claim that bone tissue has been also analyzed, but these data are not presented in the manuscript;   

Reply #4: We thank you for this suggestion, and we are pleased to say that our collaborators are currently preparing a manuscript assessing ARV impact on bone tissue, using the animal model that we present here. We have removed this mention from our manuscript since the referenced assessment of bone tissue has not yet been published.  

Comment #5: Since there are so many abbreviations, the manuscript should have a list of Abbreviations.   

Reply #5: We thank the reviewer for this suggestion, and have added a separate list of abbreviations to the manuscript (page 11, highlighted in gray before author contributions). 

Comment #6: Graph axes need to be better explained, i.e., ng/mg tissue (indicate on Y axis to what ng units refer to); Also, in figure 1, panel A, indicate what Y and X represents;   

Reply #6: We apologize and agree that axes require more detail. We have updated accordingly; please find figure 1 to be updated with axes titles (page 6).  

Comment #7: In the Results section, the first subtitle is confusing…..trafficking of human cells to mouse cells. Try re-phrase it.  

Reply #7: We apologize for the confusion and have rephrased this section to accurately reflect the data shown to “Human PBMC trafficking to peripheral tissues correlates with viral burden” Page 5 highlighted in gray, and Figure 1 legend page 6. 

Comment #8: Minors concerns: correct misspellings throughout the text (lines 85, humans instead of human; line 211, etc). 

Reply #8: We apologize for the oversight and thank the reviewer for their keen observations. These errors have been corrected.

References

  1. Hickey, W.F. Migration of hematogenous cells through the blood-brain barrier and the initiation of CNS inflammation. Brain pathology (Zurich, Switzerland) 1, 97-105 (1991).
  2. Louveau, A., et al. Structural and functional features of central nervous system lymphatic vessels. Nature 523, 337-341 (2015).

Round 2

Reviewer 2 Report

This is to let you know that I have endorsed the revised manuscript for publication.  Regards,  

Teodor-D. Brumeanu, MD.